# Assessment of Patients’ Quality of Life during Conservative Treatment after Distal Radius Fracture

**DOI:** 10.3390/ijerph192214758

**Published:** 2022-11-10

**Authors:** Piotr Ratajczak, Paweł Meller, Dorota Kopciuch, Anna Paczkowska, Tomasz Zaprutko, Krzysztof Kus

**Affiliations:** Department of Pharmacoeconomics and Social Pharmacy, Poznan University of Medical Sciences, Rokietnicka 7 Str., 60-806 Poznan, Poland

**Keywords:** bone fractures, distal radial epiphysis, Patient-Rated Wrist Evaluation (PRWE), quality of life assessment, SF-36

## Abstract

**Introduction:** This study aimed to assess patients’ quality of life after distal radius fracture treatment (at least six months, but no more than ten years, after the treatment) based on the analysis of objective and subjective parameters and the influence of the fractured side on the final results. **Materials and Methods:** The study sample consisted of 30 women who claimed to be right-handed, divided depending on the side of the fracture (left vs. right limb). Patients were evaluated with a goniometer for active wrist movement, pronation, and supination in the elbow joint. Furthermore, the global grip strength of the upper limb was assessed using a dynamometer (Biometrics Ltd.) device, after which patients were asked to complete a wrist evaluation questionnaire (PRWE) and the Polish version of the SF-36 questionnaire assessing the quality of life. **Results:** There were statistically significant differences in the active movement of the wrist of the injured limb compared to the non-injured limb. In addition, inferior results were reported for injury of the right limb to those of the left. **Conclusions:** Injury of the right limb as opposed to injury of the left limb can have a negative impact on the assessment of quality of life in patients with right-limb dominance.

## 1. Introduction

According to source data, distal radius fractures (DRFs) constitute approximately 17% of all fractures for which medical aid is provided [1,2,3]. Because of that, this injury is the most common upper limb fracture [2]. The main reason for this fracture is believed to be low-impact falls, usually on the non-dominant upper limb [2,4]. DRFs are some of the most frequently treated fractures at first aid centres. In 67–81% of all DRF cases [5] people typically fall from standing height, usually onto an extended arm, following tripping on an obstacle [2].

According to references [2,5], the issue of DRFs primarily affects the elderly and children. In the paediatric group, boys suffered from fractures more frequently than girls [5]. Early in the 21st century (2003–2005), Polish researchers (referring to earlier Swedish research) proved that these fractures were more common in young men than young women [5]. Their study failed to present the exact factors affecting the results, but the researchers assumed that higher physical activity among men is essential here. With age, differences between sexes in DRFs begin to change to increased fractures among women [5].

Further research papers show that females between fifty and sixty years of age prevail over males (of the same age) at a 2:1 ratio, but in even older groups that ratio is as high as 7:1 [5]. The authors correlate the results of their papers with menopause in women, pointing out the skeletal deficiencies associated with that process. Progressing osteoporotic processes are also responsible for increased fracture rates (growing with the patients’ age) [6,7]. The studies mentioned above also showed a gap in left- vs. right-side injuries [5]. The authors have not yet presented the final explanation of this phenomenon; however, it is assumed that the reasons lie in the proneness to defensive positions on the non-dominant side. It is also noteworthy that the dominant limb is more developed, with a more massive skeleton; hence, injuries caused by falls—which would lead to a fracture on the other hand—do not end in a fracture on this hand [5].

The incidence of this fracture may suggest that there are many suitable treatment methods and the injury itself is well-known. Nevertheless, because of the large number of biostructures within the damaged area (including forearm and wrist bones with joints and ligaments), each should be analysed separately since the treatment applied may lead to various outcomes, depending on the patient’s and the fracture’s characteristics [2]. Accurate diagnosis is key to applying an adequate treatment method. Finding the precise diagnosis can be facilitated by classification systems helping precisely match the nature of the fracture to the existing models presented in typology [1].

As mentioned above, diagnostic and therapeutic procedures help determine whether the case requires conservative treatment or surgery to reposition the fractured bones. This procedure also specifies the method (e.g., putting in plaster) and duration of limb immobilisation—all of these factors affect the outcome, and duly performed, help the bones to heal properly and in an anatomic position [2]. Commonly used therapeutic methods are:○closed reduction of bone fragments and percutaneous fixation of the fracture with Kirschner wires (based on fracture repositioning and subsequent stabilisation with percutaneously fixated Kirschner wires);○closed reduction of bone fragments with an extra external fixator (similar to the previous method, it involves repositioning and then stabilisation with an external metal structure, removed once the fracture is joined and healed);○open reduction of bone fragments with an extra palmar or dorsal locking plate (the only method involving an open reduction of bone fragments and then fixing them with an angle plate) [2,3].

However, in the future, the application of 3D patient-specific surgical guides might improve the outcome of distal radius fractures [8].

This study aimed to evaluate patients’ quality of life following conservative distal radius fracture treatment. The analysis was based on objective active range of motion and grip strength parameters as well as the results of wrist function evaluation conducted with the PRWE questionnaire. Quality of life was assessed using the SF-36 questionnaire. Analysis of the effect of the fractured side on the final results was also included. The study could answer the question of patients’ quality of life during conservative treatment after distal radius fracture.

## 2. Materials and Methods

### 2.1. Study Information

This research has been approved by the Institutional Review Board and Ethics Committee of the authors’ affiliated institutions and was conducted with patients of the Traumatology, Orthopaedics, and Hand Surgery ward of the Wiktor Dega Hospital of the Poznan University of Medical Sciences. The study (gathering results) was conducted on 21 February, 22 February, and 27 February 2019.

### 2.2. Inclusion/Exclusion Criteria

Based on interviews and analysis of the medical records shared, patient exclusion (inclusion) criteria were presented:○conservative treatment of a distal radius fracture (including patients without such a treatment),○additional fractures of upper limbs in the past and following this specific injury (including patients without other fractures),○fractures of both upper limbs (including patients with only one upper limb fracture),○dominant left upper limb (including only right-handed patients),○age under 50 and above 80 at the time of the study (including patients in the range 50–80 years old),○less than 6 months or more than 10 years between the end of treatment and the start of the study (including patients in that period).

No specific injury patterns (distal radius injuries, natures, and types), surgical management, or rehabilitation protocol were included in the study. There were 30 women who had completed DRF surgical treatment who were qualified for the study. The qualified patients were divided into two groups based on the injured limb (Table 1). All of them declared having a dominant right upper limb.

The study group of patients was further divided into two subgroups in terms of the injured limb:○Group 1—patients with left-limb fracture,○Group 2—patients with right-limb fracture.

Results for injured limb (I) versus non-injured limb (NI) were collected separately for the two groups.

### 2.3. Measurement of Objective Parameters

#### 2.3.1. Active Range of Motion

Active ranges of motion were measured using a goniometer (baseline) for each patient using the same methodology. Values established by the American Academy of Orthopedic Surgeons (AAOS) were used as standard values when measuring the upper limb motion ranges [9].

#### 2.3.2. Grip Strength

Upper-limb global grip strength was measured using a dynamometer device (Biometrics Ltd., Ladysmith, WI, USA). The test was conducted for every patient using the same method [10]. The investigator would make sure that every activity was performed correctly. For standard evaluation of peak grip strength, the device would take three tests for a patient, giving their value in kilograms and presenting an averaged result along with the coefficient of variation [10].

### 2.4. Measurement of Subjective Parameters

#### 2.4.1. Patient-Rated Wrist Evaluation (PRWE)

This questionnaire consists of two main sections, the first one with five questions on the intensity of pain felt, and the second one with ten questions on the deterioration of wrist function during the listed activities (6 questions) and daily life (4 questions) [11]. The questionnaire was handed to the patient, who assessed the pain intensity or function limitations on a scale from 0 to 10, where 0 stood for no pain or no limitation, and 10 was the highest intensity of pain or full limitation of functioning [12]. The final result of the PRWE questionnaire was calculated by summing up the results of the first section and the results of the second section divided by 2, which eventually makes a score between 0 and 100, where 0 is a total absence of pain or limited function, and 100 the highest pain intensity possible and limited functions [11].

#### 2.4.2. SF-36

The patient’s quality of life was measured using the SF-36 survey, which is a patient-reported survey of the quality of life, based on 11 questions making up 8 indicators of the physical domain—physical functioning (maximum score 50), physical role functioning (20), bodily pain (9), general health perception (24), and the mental domain—vitality (20), social role functioning (8), emotional role functioning (15), and mental health (25) [12].

The sum of the indicators for the physical domain score is a maximum 103 points, while the maximum score for the mental domain is 68. The global quality of life index has a maximum score of 171 points, but the results should be interpreted with the patient’s lower score, meaning a better quality of life [12]. This study used the Polish version of the SF-36 questionnaire [12,13].

### 2.5. Statistical Analysis

Statistical significance was established at *p* < 0.05. This paper also analysed the variable distribution normality using the Shapiro–Wilk test for deviations from a Gaussian distribution. The Wilcoxon test was performed for dependent data with a non-parametric distribution, while for data with a parametric distribution, the Student t-test was used. The mean and standard deviation (SD) functions featured in the standard Microsoft Excel package (Microsoft Corporation, Redmond, WA, USA) and Statistica 13.3 (TOBCO Software, Palo Alto, Santa Clara, CA, USA) were also used for statistical analysis.

## 3. Results

### 3.1. Active Range of Motion

#### 3.1.1. Palmar Flexion in the Sagittal Plane

The Shapiro–Wilk test determined the type of variable distribution. The Wilcoxon non-parametric test was used to analyse Group 1, and the Student t parametric test examined Group 2. Mean values and standard deviation were calculated. In both cases, injured limb (I) vs. non-injured limb (NI) results showed statistically significant differences (Table 2).

#### 3.1.2. Dorsal Flexion in the Sagittal Plane

The Shapiro–Wilk test determined the type of variable distribution. The Student t parametric test was used to analyse both Group 1 and Group 2. Mean and standard deviation were calculated. In both cases, I vs. NI results showed statistically significant differences (Table 3).

#### 3.1.3. Radial Abduction in the Frontal Plane

The Shapiro–Wilk test determined the type of variable distribution. The Wilcoxon non-parametric test was used to analyse Group 1, and the Student t parametric test to examine Group 2. Mean and standard deviation were calculated. In both cases, I vs. NI results showed statistically significant differences (Table 4).

#### 3.1.4. Ulnar Abduction in the Frontal Plane

The Shapiro–Wilk test determined the type of variable distribution. The Student t parametric test was used to analyse both Group 1 and Group 2. Mean and standard deviation were calculated. In both cases, I vs. NI results showed statistically significant differences (Table 5).

#### 3.1.5. Forearm Pronation

The Shapiro–Wilk test determined the type of variable distribution. The Wilcoxon non-parametric test was used to analyse both Group 1 and Group 2. Mean and standard deviation were calculated. In both cases, I vs. NI results showed statistically significant differences (Table 6).

#### 3.1.6. Forearm Supination

The Shapiro–Wilk test determined the type of variable distribution. The Student t parametric test was used to analyse both Group 1 and Group 2. Mean and standard deviation were calculated. In both cases, I vs. NI results showed statistically significant differences (Table 7).

### 3.2. Upper Limb Global Grip Strength

#### 3.2.1. Maximum Grip Strength

The Shapiro–Wilk test determined the type of variable distribution. The Student t parametric test was used to analyse both Group 1 and Group 2. Mean and standard deviation were calculated. In both cases, I vs. NI results showed statistically significant differences (Table 8).

#### 3.2.2. Mean of Three Results for Grip Strength

The Shapiro–Wilk test determined the type of variable distribution. The Student t parametric test was used to analyse both Group 1 and Group 2. Mean and standard deviation were calculated. In both cases, I vs. NI results showed statistically significant differences (Table 9).

### 3.3. Patient-Reported Wrist Function Evaluation

The results of both groups were analysed using the methodology applicable to interpreting the PRWE questionnaire. The mean result with standard deviation was determined for each of the groups, divided into global and partial results for two categories: “pain” and “function” (Table 10).

### 3.4. Quality of Life Evaluation

The results of both groups were analysed using the methodology applicable to interpreting the Polish version of the SF-36 survey. The percentage mean outcome was determined for each of the eight quality of life indexes, for both life domains, and the overall quality of life index (Table 11).

## 4. Discussion

This study’s main findings were related to active range of motion, grip strength, wrist function evaluation, and quality of life assessments for both groups based on injured and non-injured sides. The study population included women who had undergone surgery for DRF. According to studies by other authors, the surgery method as such does not have a significant effect on the treatment outcome [14], therefore this factor was not taken into account while performing the analysis.

The obtained results show a trend of inferior outcomes in the injured limb compared to the non-injured limb despite the progress in time after the injury. Interestingly, this correlation was observed in the group with an injured dominant (right) limb. It should be strongly repeated that all the patients included in the study declared themselves right-handed. Objective measurement parameters of the active range of motion yielded mean results indicating restoration of a fuller range of motion in the carpal joint of the left injured limb than in the right injured limb for palmar flexion, with an 8.8% difference between the limbs (Table 2); dorsal flexion, with a 10.2% difference between the limbs (Table 3); radial abduction, with a 1% difference (Table 4); and ulnar abduction, with a 1.7% difference (Table 5). Moreover, results for the injured limb compared to the non-injured limb (regardless of whether it was the right or left hand) in all of the above ranges were significantly different. The study also found restoration of a fuller range of motion in the elbow joint and strength in the right injured limb than in the left injured limb for forearm pronation, with a 3.1% difference (Table 6); forearm supination, with a 2.2% difference (Table 7); maximum grip strength, with a 2.8% difference (Table 8); and the average of three results of grip strength measurement, with a 0.5% difference (Table 9). For the above ranges, only the group with fractured left limbs had statistically significant differences in the analysed parameters. The group with fractured right limbs had statistically significant differences only in forearm supination. Gerald Gruber et al. [15] followed up 54 patients who had undergone surgery with angle plate fixation for 6 years and eventually reported averaged results which corroborate with our results, showing that the injured limb had inferior averaged results compared to the non-injured limb in terms of active range of motion and handgrip strength. On the other hand, Gianluca Testa et al. [14] evaluated 39 patients who had been at least 65 years old at the time of fracture and had undergone surgery 6 months after the surgery was completed. The results of these researchers also corroborate our results regarding the active range of motion.

In the case of DRFs, DASH is the questionnaire more frequently used in other references to evaluate pathology in the upper limb (including limited function and pain and pain-related inconveniences of daily life) instead of the PRWE questionnaire used in this study [16]. Nevertheless, a research paper from 2018 presents both questionnaires as offering similar efficacy (in evaluating limb disability), and both are assessed as highly effective [17]. Averaged PRWE questionnaire results for the groups in the present study indicated limited function and more significant pain in patients with left-limb injury than in those with right-limb injury. Group 1 (with a fractured left limb) scored 18.8 points out of 100 for limited functioning and 9.8 out of 50 for pain, while Group 2 (with a fractured right limb) scored 16.8 points for limited functioning and 8.8 for pain (Table 10). In both cases, the difference was slight and may have been caused by the patient’s interpretation, since PRWE’s methodology involves an option to leave specific questions unanswered [11]. In both cases, the lower the result, the smaller the intensity of pain complaints and the fuller the limb function. In our opinion, the lower results for Group 2 regarding pain could be related to the fact that dominant limb rehabilitation (every patient in the study was right-handed) is more fluid due to muscle memory and better motor control of the dominant hand. That is why patients could achieve a higher (and faster) pain reduction. On the other hand, patients with an injured left hand presented higher values for the restoration of the range of motion, with simultaneously observed greater pain. This phenomenon could also be linked to psychological aspects, especially when we directly compare the pain results to the restoration of the range regarding palmar flexion, dorsal flexion, radial abduction, and ulnar abduction (Table 2, Table 3, Table 4 and Table 5).

The results are similar to those published in April 2019 by Roderick van Leerdam et al. [18], who studied 285 patients, 82 of whom had undergone surgery, for 3.8 years. For functional evaluation of the limb, they used the PRWE questionnaire, dividing the patients into two main groups in terms of the treatment method and subgroups in terms of AO classification. Averaged results turned out to be correlated both with the method of treatment and type of fracture: the “Type A” group (extra-articular fracture) of 14 patients had an average score of 20 points; the “Type B” group (intra-articular fracture with fracture cleft in the sagittal plane) of 19 patients had an average score of 25 points; and the “Type C” group (intra-articular fracture with fracture cleft in the frontal plane) of 54 patients had an average score of 13 [18].

This study was also intended to evaluate the quality of life of DRF patients who completed their rehabilitation stage, i.e., to determine the effect of the fracture on the patient’s everyday life, limited functioning, and mental health. The summed-up and averaged results according to the SF-36 survey methodology tend to show higher (i.e., more deficient according to the method) results in the right limb fracture group. They are particularly notable in the social functioning aspect (a 19.5% difference—Table 11) and physical and emotional role functioning limitations (a difference of 6% and 33.2%). The final quality of life index along with the physical and mental domains indicate that on average, patients with a left-hand DRF evaluate their quality of life higher than those with a right-hand fracture (Table 11), which may seem logical considering the impact of the dominant limb on our everyday life [19]. Beaulé Paul-Emile [19] investigated and confirmed that patients with DRF of the dominant limb reported a broader range of hand function impairment and more intense pain than patients with a non-dominant hand fracture. However, the authors of that study indicated a higher potential for improvement of the dominant limb, as opposed to the results of our study [19]. Golec et al. [20] evaluated the quality of life of 71 patients of both sexes and various ages using the SF-36 survey at the following times: 7 days, 6 weeks, 3 months, and 6 months following the fracture. Their results corroborate those obtained in the present study. The patients would have lower scores in the quality of life survey if the fracture affected the dominant limb [20].

The results obtained in this study were also corroborated by previous studies, showing a particular problem in restoring the function in the injured limb even years after the fracture. However, no more-extensive studies were found evaluating the results of dominant limb vs. non-dominant limb fractures. The references and the previously quoted papers only mentioned that the dominant limb had a higher probability of improving the outcome, which is inconsistent with our results [15,19]. Initially, this study assumed that the dominant limb performed most manual actions. However, the results show an average more-limited mobility of the right injured limb and greater strength of the non-dominant left limb than the injured (dominant) left limb. This phenomenon may be due to the necessity of using the non-dominant limb throughout the treatment and immobilisation phase, which helped improve its function [3].

The present study has certain limitations. Both groups included patients who did not complain of pain, were not bothered by certain mobility limitations, and whose quality of life depended on their current health complaints. We were trying to create a homogenous group of patients for reliable statistical analysis. This has not been easy for DRF, since the final treatment outcome depends on multiple factors; therefore, the results should be confirmed in further, more extensive studies, combining data of patients (matching the imposed diagnostic framework) from several hospitals.

## 5. Conclusions

It must be concluded that there were statistically significant differences in the active movement of the wrist of the injured limb compared to the non-injured limb. In addition, inferior results were reported for injury to the right (dominant) limb than to the left. Moreover, injury to the right limb can negatively impact the assessment of the quality of life in patients with right-limb dominance.

## Figures and Tables

**Table 1 ijerph-19-14758-t001:** Characteristics of the study group.

Study Group	Population	Mean Age and Standard Deviation (SD)	Average Number of Years Following the Surgery and SD
Total	30	66 ± 7	5 ± 3 years
Women with left upper limb fracture (Group 1)	17	66 ± 7	5.25 ± 3.5 years
Women with right upper limb fracture (Group 2)	13	66 ± 7	4.75 ± 2.75 years

**Table 2 ijerph-19-14758-t002:** Comparison of results for palmar flexion.

Group	Variable	Hand	Mean with SD	T-1 vs. T-2	Significance Level
1	I palmar flexion	L	56.7° ± 10.5° ***	8.8% The group with a traumatic left hand achieved higher results	0.0005
	NI palmar flexion	R	66.6° ± 7.6°		
2	I palmar flexion	R	51.7° ± 10.2° *		0.0091
	NI palmar flexion	L	58.8° ± 7°		

* Statistically significant *p* < 0.05 I vs. NI; I—injured limb; NI—non-injured limb.

**Table 3 ijerph-19-14758-t003:** Comparison of results for dorsal flexion.

Group	Variable	Hand	Mean with SD	T-1 vs. T-2	Significance Level
1	I dorsal flexion	L	49.9° ± 8.2° ***	10.2% The group with a traumatic left hand achieved higher results	0.0001
	NI dorsal flexion	R	57.6° ± 7.9°		
2	I dorsal flexion	R	44.8° ± 13.2° ***		0.0071
	NI dorsal flexion	L	55.2° ± 6.2°		

* Statistically significant *p* < 0.05 I vs. NI; I—injured limb; NI—non-injured limb.

**Table 4 ijerph-19-14758-t004:** Comparison of results for radial abduction.

Group	Variable	Hand	Mean with SD	T-1 vs. T-2	Significance Level
1	I radial abduction	L	19.1° ± 3.4° ***	1% The group with a traumatic left hand achieved higher results	0.0001
	NI radial abduction	R	23.6° ± 3.3°		
2	I radial abduction	R	18.9° ± 5.2° ***		0.0071
	NI radial abduction	L	22.5° ± 4.4°		

* Statistically significant *p* < 0.05 I vs. NI; I—injured limb; NI—non-injured limb.

**Table 5 ijerph-19-14758-t005:** Comparison of results for ulnar abduction.

Group	Variable	Hand	Mean with SD	T-1 vs. T-2	Significance Level
1	I ulnar abduction	L	23.8° ± 8.2° ***	1.7% The group with a traumatic left hand achieved higher results	0.0001
	NI ulnar abduction	R	31.1° ± 5.5°		
2	I ulnar abduction	R	23.4° ± 7° ***		0.0054
	NI ulnar abduction	L	30.1° ± 7.1°		

* Statistically significant *p* < 0.05 I vs. NI; I—injured limb; NI—non-injured limb.

**Table 6 ijerph-19-14758-t006:** Comparison of results for forearm pronation.

Group	Variable	Hand	Mean with SD	T-1 vs. T-2	Significance Level
1	I forearm pronation	L	83.4° ± 6.5° ***	3.1% The group with a traumatic right hand achieved higher results	0.0117
	NI forearm pronation	R	86.7° ± 4.6°		
2	I forearm pronation	R	86.1° ± 4.6°		0.1422
	NI forearm pronation	L	87.9° ± 3.5°		

* Statistically significant *p* < 0.05 I vs. N; I—injured limb; NI—non-injured limb.

**Table 7 ijerph-19-14758-t007:** Comparison of results for forearm supination.

Group	Variable	Hand	Mean with SD	T-1 vs. T-2	Significance Level
1	I forearm supination	L	78.8° ± 5.8° ***	2.2% The group with a traumatic right hand achieved higher results	0.005
	NI forearm supination	R	83.1° ± 4.3°		
2	I forearm supination	R	80.6° ± 4.8° ***		0.0243
	NI forearm supination	L	83.4° ± 3.8°		

* Statistically significant *p* < 0.05 I vs. NI; I—injured limb; NI—non-injured limb.

**Table 8 ijerph-19-14758-t008:** Comparison of results for maximum strength.

Group	Variable	Hand	Mean with SD	T-1 vs. T-2	Significance Level
1	I maximum strength	L	21.1 kg ± 2.5 kg ***	2.8% The group with a traumatic right hand achieved higher results	0.0001
	NI maximum strength	R	24 kg ± 2 kg		
2	I maximum strength	R	21.7 kg ± 3.6 kg		0.1026
	NI maximum strength	L	23.3 kg ± 2.5 kg		

* Statistically significant *p* < 0.05 I vs. NI; I—injured limb; NI—non-injured limb.

**Table 9 ijerph-19-14758-t009:** Comparison of results for the mean value of three grip strength measurements.

Group	Variable	Hand	Mean with SD	T-1 vs. T-2	Significance Level
1	I mean of three measurements	L	20.2 kg ± 2.3 kg ***	0.5% The group with a traumatic right hand achieved higher results	0.0001
	NI mean of three measurements	R	23.1 kg ± 2.1 kg		
2	I mean of three measurements	R	20.3 kg ± 3.4 kg		0.0845
	NI mean of three measurements	L	22 kg ± 2.6 kg		

* Statistically significant *p* < 0.05 I vs. NI; I—injured limb; NI—non-injured limb.

**Table 10 ijerph-19-14758-t010:** Comparison of results of the PRWE questionnaire.

Group	Possible Score	Mean with SD
1	Total PRWE score—100 points possible	18.6 ± 18.6
	Pain subscale—50 points possible	9.8 ± 9.1
	Functional subscale—50 points possible	8.8 ± 9.9
2	Total PRWE score—100 points possible	16.8 ± 25.7
	Pain subscale—50 points possible	8.8 ± 12.5
	Functional subscale—50 points possible	8 ± 13.6

**Table 11 ijerph-19-14758-t011:** Quality of life index.

PHYSICAL DOMAIN				
Group	Physical Functioning	Physical Role Functioning	Bodily Pain	General Health
1	21.8%	38.2%	36.6%	58.3%
2	25.4%	44.2%	39.3%	60.3%
**MENTAL DOMAIN**				
**Group**	**Vitality**	**Social functioning**	**Emotional role functioning**	**Mental health**
1	54.4%	19.9%	7.8%	39.1%
2	54.6%	39.4%	41%	42.8%
**OVERALL INDEX**				
**Group**	**Physical health dimension**	**Mental health dimension**	**Quality of life index**	
1	34.8%	34.4%	34.7%	
2	38.4%	44.3%	40.8%	

## Data Availability

The datasets analysed during the current study are available from the corresponding author upon reasonable request.

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
