# Peer review of "Assessment of Patients’ Quality of Life during Conservative Treatment after Distal Radius Fracture"

_ijerph, 2022, doi:10.3390/ijerph192214758_

Round 1
Reviewer 1 Report
please see attached file

Reviewer 2 Report
1. I would suggest that authors should mention patient were included as conservative treatment of a distal radius fracture (i.e., not with surgery) both in tile and in introduction section.
2. I was surprised that the Mean age and standard deviation (SD) were same in Group1 and 2.
3. It would be better included general healthy patient/ without hand/wrist injury as control group.
Reviewer 3 Report
The article evaluates functional outcomes after surgical treatment of patients with distal radius fracture in right-handed patients. In particular, it compares differences according to the side of the fracture.
There are already similar studies in the literature, however, I believe that further evidence may add knowledge to the topic. In addition, the study is methodologically well conducted and the article is well written. In particular: the introduction is comprehensive, the methods precise and well described (including statistical analysis), the results are clear, the discussion detailed, and the conclusions consistent with the results.
I have only a few suggestions for authors for minor revisions:
1) move the study aims to the end of the introduction;
2) describe the different surgical treatments used in the methods and not in the discussion (lines 223-230); also specify the number of each of these treatments for each group (right side versus left side);
3) include details regarding postoperative management in the methods: immobilization, resumption of movement, exercise, depending on the different surgical treatments;
4) include in discussions the authors' opinions on the reasons for the differences in outcome between right and left side (particularly regarding restoration of range of motion and residual pain).
Thank you.
